

# Higher-order mean-field theory of chiral waveguide QED

Kasper J. Kusmierek[1], Sahand Mahmoodian[1,2], Martin Cordier[3],
Jakob Hinney[4,5], Arno Rauschenbeutel[3,4], Max Schemmer[3,*,∘],
Philipp Schneeweiss[3,4], Jürgen Volz[3,4] and Klemens Hammerer[1,†]

**1** Institute for Theoretical Physics, Leibniz University Hannover,
Appelstrasse 2, 30167 Hannover, Germany
**2** Centre for Engineered Quantum Systems, School of Physics,
The University of Sydney, Sydney, NSW 2006, Australian
**3** Department of Physics, Humboldt-Universität zu Berlin, 10099 Berlin, Germany
**4** Vienna Center for Quantum Science and Technology, TU Wien-Atominstitut,
Stadionallee 2, 1020 Vienna, Austria
**5** Department of Electrical Engineering, Columbia University,
New York, New York 10027, USA

⋆ Corresponding author for experiment: maximilian.schemmer@ino.cnr.it
∘ Current address: Istituto Nazionale di Ottica del Consiglio Nazionale
delle Ricerche (CNR-INO), 50019 Sesto Fiorentino, Italy
† Corresponding author for theory: klemens.hammerer@itp.uni-hannover.de

## Abstract

Waveguide QED with cold atoms provides a potent platform for the study of non-equilibrium, many-body, and open-system quantum dynamics. Even with weak coupling and strong photon loss, the collective enhancement of light-atom interactions leads to strong correlations of photons arising in transmission, as shown in recent experiments. Here we apply an improved mean-field theory based on higher-order cumulant expansions to describe the experimentally relevant, but theoretically elusive, regime of weak coupling and strong driving of large ensembles. We determine the transmitted power, squeezing spectra and the degree of second-order coherence, and systematically check the convergence of the results by comparing expansions that truncate cumulants of few-particle correlations at increasing order. This reveals the important role of many-body and long-range correlations between atoms in steady state. Our approach allows to quantify the trade-off between anti-bunching and output power in previously inaccessible parameter regimes. Calculated squeezing spectra show good agreement with measured data, as we present here.

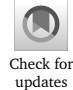

# 1 Introduction

The strong and tunable interactions among photons and atoms achievable in engineered nanophotonic structures present exciting prospects for fundamental studies in non-equilibrium many-body physics and for applications in quantum technology [1–3]. Waveguide QED [4–8], specifically, offers unique opportunities to study the propagation of light in highly nonlinear media and in the realm of collective coupling with atoms [9–17]. A distinctive feature of QED with nanophotonic waveguides is the possibility of realizing a chiral light-matter interaction in which atoms couple exclusively to photons propagating unidirectionally [18–23]. It was shown that pulse propagation through an ensemble of non-interacting atoms strongly and chirally coupled to a waveguide is governed by a rich structure of multi-photon states that can lead to time-ordered many-body states of light [24, 25]. Remarkably, even in the case of weak coupling, where photons are predominantly scattered out of the waveguide, the interplay of losses with the nonlinearity of atoms results in strong correlations of the light [26]. In recent experiments, these photon-photon correlations have been demonstrated in the form of (anti)-bunching [13] and squeezing [14] in light transmitted through an ensemble of two-level systems that were weakly and chirally coupled to a waveguide.

Due to their non-equilibrium, many-body, and open-system dynamics, the theoretical description of such experiments is a major challenge at present. For lossless chiral systems, the scattering matrix can be expressed analytically even in the many-body regime [27, 28]. For the experimentally more relevant regime of weak coupling, this approach can also be applied in the subspace with few excitations, giving good agreement with measurements for small input powers [13, 14, 26]. However, the same method cannot be applied for stronger driving fields approaching saturation, where states with larger numbers of excitations contribute significantly. Instead of propagating the wave function of photons by means of an expansion on scattering eigenstates, it is also possible to infer the properties of transmitted light from the dynamics of atoms using input-output relations, and the quantum-regression theorem [29, 30]. In principle, this requires the solution of an open many-particle spin model [31], which in turn is only possible exactly in the subspace involving few excitations.

An approximate treatment of the many-body dynamics of atoms for strong driving may exploit the fact that the coupling to the waveguide is weak. Since the dominant scattering of

photons from the waveguide acts as a local decoherence channel for each atom and correlations between atoms are induced only via weak collective scattering in guided field modes, one can expect many-body correlations to be limited. Approximate descriptions for finitely correlated systems in terms of matrix-product-states (MPS) [32] have been applied to waveguide QED systems with good success [24,33]. However, because the one-dimensional geometry supports infinite-range interactions, the finite correlation length imposed by MPS may fail to capture important features, which becomes especially problematic for larger systems. Indeed, the long-range nature of interactions [12] and correlations [15] has been explored in recent studies.

Here, we employ an improved mean-field theory based on a higher order cumulant expansion [34] to solve for the dynamics of a strongly-driven, weakly-coupled, chiral chain of atoms and thus determine the photon statistics of the transmitted light. In lowest order, this expansion reduces to ordinary mean-field theory, and essentially assumes a tensor product state of atoms. While this reproduces the equations of classical electrodynamics, it obviously fails to account for the collective effects due to quantum correlations [35]. $n$-th order mean-field expansions, accounting systematically for genuine $n$-particle correlations, have received growing interest lately in the context of collective interactions of light with atomic ensembles [36–39]. In general, such an expansion reduces the effective dimensionality of the problem at the cost of introducing a nonlinearity in the equations of motion whose complexity grows with the order of expansion. Remarkably, as we show here, when applied to a chirally coupled system, the problem stays effectively linear. This avoids numerical issues usually arising at larger orders of truncation, and allows us to compare results for different expansions even for systems of considerable size. Using this method, we determine squeezing spectra and the degree of second-order coherence in parameter regimes not accessible with other methods. In the low power regime, we find results consistent with those found from the expansion in the two-photon subspace [26]. For large driving, we find that higher order correlations of atoms play a significant role for describing the correlations in transmitted light. We also study the spatial characteristics of two-particle correlations, and show that the system develops intriguing patterns of long-range correlations in steady state. Theoretical predictions for squeezing spectra are compared with measurements and show good agreement, extending the discussion in [14] to the regime of large driving powers. The predictions for the antibunching achievable in a chiral waveguide system allow a discussion of the quality, in terms of the Mandel $Q$ parameter, of a stationary single photon source envisioned in [13,40].

The paper is structured as follows: In section 2 we introduce the cascaded systems master equation governing the dynamics of the chiral system. We also introduce the correlation functions of light, which we aim to determine from the dynamics of atoms. Section 3 deals with the lowest order approximation to the system, i.e. mean-field theory. In section 4, we introduce the higher order cumulant expansion, provide a systematic comparison of expansions at various order, and discuss the role and characteristics of atomic correlations.

## 2 Cascaded system master equation

We consider the system shown in Fig. 1. An arrangement of $N$ two-level atoms is chirally coupled to a waveguide, i.e. the atoms effectively couple only to photons propagating in a single direction. In addition, each atom couples independently to environmental modes. The total rate of spontaneous emission $\Gamma$ is accordingly composed of an emission rate $\beta\Gamma$ into the waveguide and a rate $(1-\beta)\Gamma$ for decay into the environment, where $0 \leq \beta \leq 1$. A continuous-wave coherent field is coupled into the waveguide and drives the atomic transition at frequency $\omega_0$. In the following, we will focus on the case of a resonant driving field investigated also in recent experiments [13,14]. The strength of the coherent drive is characterized by its mean

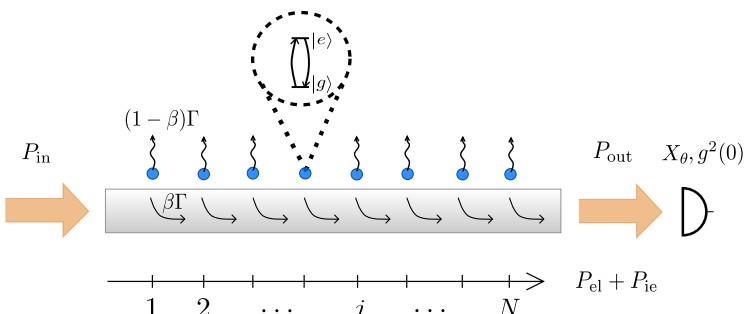

Figure 1: Schematic of setup. In an array of $N$ two-level atoms, each atom can emit either at a rate $\beta\Gamma$ chirally (unidirectionally) into a waveguide or at $(1-\beta)\Gamma$ into environmental modes. A continuous-wave coherent field resonant with the atomic transition is coupled into the waveguide with a power $P_{\text{in}}$. We determine the transmitted power $P_{\text{out}}$, its components due to elastic and inelastic scattering ($P_{\text{el}}$, $P_{\text{ie}}$), the squeezing spectra of light quadratures $X_\theta$, and the degree of second order coherence $g^2(0)$.

photon flux $P_{\text{in}}$ corresponding to an input power $\hbar\omega_0 P_{\text{in}}$. For the sake of simplicity, we will not distinguish these quantities and refer also to $P_{\text{in}}$ and related quantities as powers. In the following, we are interested in the regime of strong drive, by which we understand here an input power $P_{\text{in}}$ approaching the saturation power $P_{\text{sat}} = \Gamma/\beta$.

In this arrangement, the reduced state $\rho_N$ of the $N$ atoms (after elimination of the photon field in standard Born-Markov approximation) evolves according to a cascaded system master equation [23,29,30,41]

$$\frac{1}{\Gamma}\frac{d\rho_N}{dt} = -i\sum_{j=1}^{N}\sqrt{\frac{P_{\text{in}}}{P_{\text{sat}}}}\big[\sigma_j^- + \sigma_j^+, \rho_N\big] + (1-\beta)\sum_{j=1}^{N}D\big[\sigma_j^-\big]\rho_N$$

$$+ \frac{\beta}{2}\sum_{\substack{j,l=1 \\ j>l}}^{N}\big[\sigma_l^+\sigma_j^- - \sigma_j^+\sigma_l^-, \rho_N\big] + \beta\, D\Big[\sum_{j=1}^{N}\sigma_j^-\Big]\rho_N =: L_N\rho_N. \quad (1)$$

The master equation is written in a frame rotating at $\omega_0$. The first two terms on the right hand side describe, respectively, the coherent resonant drive and decay of atoms to ambient modes. These two processes affect each of the atoms independently. $D[x]\rho = x\rho x^\dagger - \frac{1}{2}(x^\dagger x\rho + \rho x^\dagger x)$ denotes a Lindblad term with jump operator $x$. The cascaded (chiral) coupling of atoms to the waveguide is accounted for by the two terms in the last line. These describe collective dynamics and induce correlations among atoms.

The field coupled out of the waveguide is described by the annihilation operator $a_{\text{out}}(t)$ which fulfills the input-output relation

$$a_{\text{out}}(t) = a_{\text{in}}(t) + \sqrt{P_{\text{in}}} - i\sqrt{\beta\Gamma}\sum_{i=1}^{N}\sigma_i^-(t). \quad (2)$$

The first two terms on the right hand side are, respectively, vacuum noise and coherent amplitude (assumed to be real valued) of the input field. The last term represents the field radiated by the atomic dipoles, and accounts for all effects arising from scattering of photons on atoms (including damping of the coherent amplitude). The atomic operators appear as a collective lowering operator as the detection of a photon exiting the waveguide can be emitted from any of the emitters. The vacuum noise on the right hand side of Eq. (2) will be suppressed in

the following since we will be exclusively interested in normally ordered quantities to which vacuum fluctuations do not contribute.

Specifically, we want to characterize the properties of the transmitted light while varying the input power $P_{\mathrm{in}}/P_{\mathrm{sat}}$, the atomic waveguide coupling $\beta$, and the number of emitters $N$. A crucial parameter in this system is its optical depth $OD = 4\beta N$. We are interested in the transmitted power $P_{\mathrm{out}} = \langle a_{\mathrm{out}}^\dagger a_{\mathrm{out}} \rangle = P_{\mathrm{el}} + P_{\mathrm{ie}}$, its components due elastically and inelastically scattered photons $P_{\mathrm{el}}$ and $P_{\mathrm{ie}}$, the squeezing properties of the quadratures of the transmitted field, and its second-order coherence, i.e. the (anti)bunching of the transmitted photons. Via the input-output relation (2), these quantities ultimately relate to correlation functions of atomic observables. In particular, we have

$$P_{\mathrm{el}} = |\langle a_{\mathrm{out}} \rangle|^2 = \left| \sqrt{P_{\mathrm{in}}} - i\sqrt{\beta\Gamma} \sum_{i=1}^{N} \langle \sigma_i^-(t) \rangle \right|^2 , \tag{3}$$

$$P_{\mathrm{ie}} = \langle a_{\mathrm{out}}^\dagger a_{\mathrm{out}} \rangle - |\langle a_{\mathrm{out}} \rangle|^2 = \beta\Gamma \sum_{i,j=1}^{N} \langle\!\langle \sigma_i^+(t) \sigma_j^-(t) \rangle\!\rangle . \tag{4}$$

$P_{\mathrm{el}}$ corresponds to the power due to the interference of the coherent input and the field radiated by the average atomic dipoles. $P_{\mathrm{ie}}$ is the power radiated from dipole fluctuations as described by their covariance or (second-order) cumulant $\langle\!\langle \sigma_i^+ \sigma_j^- \rangle\!\rangle$, which is generally defined by

$$\langle\!\langle AB \rangle\!\rangle = \langle AB \rangle - \langle A \rangle \langle B \rangle . \tag{5}$$

While Eqs. (3) and (4) apply for arbitrary time, we focus in the following exclusively on the stationary dynamics of the system, implicitly taking a long-time limit. Furthermore, the squeezing spectrum of light quadratures $X_\theta(t) = \frac{1}{2}\left( a_{\mathrm{out}}(t) e^{i\theta} + \mathrm{h.c.} \right)$ is quantified by the spectral density

$$:S_\theta(\omega): = \int_0^\infty d\tau \left( e^{i\omega\tau} + e^{-i\omega\tau} \right) \langle :\delta X_\theta(\tau)\delta X_\theta(0): \rangle \tag{6}$$

of quadrature fluctuations $\delta X_\theta(t) = X_\theta(t) - \langle X_\theta(t) \rangle$. The colons in $:A:$ denote normal and time ordering of $A$. The angle $\theta$ is the local oscillator phase with respect to the coherent drive. With the input-output-relation (2) the two-time correlations of quadrature fluctuations can be related to those of atomic operators,

$$\langle :\delta X_\theta(\tau)\delta X_\theta(0): \rangle = \frac{\beta\Gamma}{4} \sum_{i,j=1}^{N} \langle\!\langle \sigma_j^+(0)\sigma_i^-(\tau) \rangle\!\rangle - e^{2i\theta} \langle\!\langle \sigma_i^-(\tau)\sigma_j^-(0) \rangle\!\rangle + \mathrm{h.c.} \tag{7}$$

The squeezing spectra in Eq. (6) with regard to two conjugate quadratures, say $\theta = 0, \pi/2$, can be combined to determine the spectrum of inelastically scattered photons,

$$S_{\mathrm{ie}}(\omega) = \int_{-\infty}^{\infty} d\tau\, e^{-i\omega\tau} \langle\!\langle a_{\mathrm{out}}^\dagger(\tau) a_{\mathrm{out}}(0) \rangle\!\rangle = :S_0(\omega): + :S_{\pi/2}(\omega): . \tag{8}$$

The frequency integral of this spectrum in turn corresponds to the power in the inelastically scattered field in Eq. (4),

$$P_{\mathrm{ie}} = \frac{1}{2\pi} \int_{-\infty}^{\infty} d\omega\, S_{\mathrm{ie}}(\omega) = \langle :\delta X_0(0)\delta X_0(0): \rangle + \langle :\delta X_{\pi/2}(0)\delta X_{\pi/2}(0): \rangle , \tag{9}$$

where we introduced the integrated quadrature fluctuations,

$$\langle :\delta X_\theta(0)\delta X_\theta(0): \rangle = \frac{1}{2\pi} \int_{-\infty}^{\infty} d\omega \ :S_\theta(\omega): . \tag{10}$$

Finally, we will also explore the normalized second order correlation function of the output field at equal times,

$$g^{(2)}(0) = \frac{\left\langle a_{\text{out}}^{\dagger}(0) a_{\text{out}}^{\dagger}(0) a_{\text{out}}(0) a_{\text{out}}(0) \right\rangle}{P_{\text{out}}^2} . \tag{11}$$

Its expression in terms of atomic operators follows from the direct application of the input-output relation in Eq. (2), but turns out to be rather lengthy and is therefore moved to Eq. (A.5) in Appendix A. Evidently, $g^{(2)}(0)$ involves atomic moments among up to four atoms.

In summary, all quantities of interest ultimately depend on mean values and two-body, as well as higher order, correlations of atomic observables. These can, in principle, be calculated from the master equation (1), in combination with the quantum regression theorem as necessary. However, owing to the exponential scaling of the dimension of $\rho_N$ in the number of atoms and due to the correlations that arise between them during collective decay through the waveguide an exact solution is unfeasible. This is the case even in the region of weak coupling to the waveguide, $\beta \ll 1$, when the optical depth is large $OD = 4\beta N > 1$. However, in this regime the atoms mostly decay non-collectively, and it is therefore to be expected that correlations between atoms remain weak, at least in the sense that many-particle correlations are less pronounced than correlations between few particles. On this basis, we construct in the following approximate solutions of the steady state of the master equation in a mean-field approach and, in systematic extensions of this, in higher-order cumulant expansions.

## 3 Mean-field theory

### 3.1 Transmitted power

The equations of motion for the expectation values of the $x$, $y$, and $z$ Pauli operators are

$$\frac{1}{\Gamma} \frac{d}{dt} \langle \sigma_j^x \rangle = -\frac{1}{2} \langle \sigma_j^x \rangle ,$$

$$\frac{1}{\Gamma} \frac{d}{dt} \langle \sigma_j^y \rangle = -\frac{1}{2} \langle \sigma_j^y \rangle - 2\alpha_j \langle \sigma_j^z \rangle + \beta \sum_{l=1}^{j-1} \langle\!\langle \sigma_j^z \sigma_l^y \rangle\!\rangle ,$$

$$\frac{1}{\Gamma} \frac{d}{dt} \langle \sigma_j^z \rangle = 2\alpha_j \langle \sigma_j^y \rangle - \langle \sigma_j^z \rangle - 1 - \beta \sum_{l=1}^{j-1} \left( \langle\!\langle \sigma_j^x \sigma_l^x \rangle\!\rangle + \langle\!\langle \sigma_j^y \sigma_l^y \rangle\!\rangle \right), \tag{12}$$

as follows from Eq. (1) without approximation. For simplicity, we have already omitted quantities of the form $\langle\!\langle \sigma_i^x \sigma_j^{y,z} \rangle\!\rangle$, anticipating that these vanish for resonant drive. In these equations we already expressed two-body correlations through second order cumulants using Eq. (5). In this way, an effective field driving the $j$–th atom naturally emerges,

$$\alpha_j = \alpha_1 - i\beta \sum_{l=1}^{j-1} \langle \sigma_l^- \rangle , \qquad \alpha_1 = \sqrt{\frac{P_{\text{in}}}{P_{\text{sat}}}} , \tag{13}$$

where $\alpha_1$ is the amplitude experienced by the first atom. The sum in the expression for $\alpha_j$ accounts for the field radiated coherently by all atoms to the left of the $j$–th one. For resonant driving, the expectation values of the out-of-phase atomic dipoles vanish in steady state, $\langle \sigma_j^x \rangle = 0$, and thus the effective driving field $\alpha_j = \alpha_1 - \frac{\beta}{2} \sum_{l=1}^{j-1} \langle \sigma_l^y \rangle$ is real. It is also useful to note that the output power corresponding to the elastically scattered photons in Eq. (3) can be considered as the effective driving field that would be seen by a hypothetical $(N+1)^{\text{th}}$ atom, $P_{\text{el}}/P_{\text{sat}} = \alpha_{N+1}^2$.

To close the system of Eqs. (12), it would have to be supplemented by corresponding equations for all correlations up to $N$ particles, which would amount to the exact solution of the master equation. The mean-field approach is the lowest-order approximation that yields a closed system of equations and corresponds to neglecting all second-order cumulants $\langle\!\langle \sigma_j^\mu \sigma_l^\nu \rangle\!\rangle$ in Eqs. (12) where $\mu, \nu = x, y, z$. In view of Eq. (5), this is tantamount to approximate

$$\langle AB \rangle \simeq \langle A \rangle \langle B \rangle \,, \tag{14}$$

where $A$ and $B$ are operators referring to different atoms. This approach corresponds to mean-field theory, where a product state ansatz is made for the density matrix $\rho_N = \bigotimes_i \rho^{(i)}$ with single particle states $\rho^{(i)}$. Since the system considered here is not translationally invariant, the single particle states will not be identical. In this approximation and in stationary state, Eqs. (12) are solved by

$$\langle \sigma_j^z \rangle = -\frac{1}{1+8\alpha_j^2} \,, \qquad \langle \sigma_j^y \rangle = \frac{4\alpha_j}{1+8\alpha_j^2} \,. \tag{15}$$

Substituting in Eq. (13) yields a recurrence relation for the effective driving field in mean-field theory $\alpha_j = \alpha_1 - 2\beta \sum_{l=1}^{j-1} \alpha_l/(1+8\alpha_l^2)$.

An approximate solution can be constructed by considering the difference equation $\Delta\alpha_j = \alpha_{j+1} - \alpha_j = -2\beta\alpha_j/(1+8\alpha_j^2)$. In the continuous limit (replacing the index $j$ by a continuous variable $z \in [0, N]$), the solution of the corresponding differential equation for $\alpha(z)$ yields the *Lambert law* for the elastically scattered power,

$$P_{\mathrm{el}}(z) = P_{\mathrm{sat}}\, \alpha(z)^2 = P_{\mathrm{in}} \frac{w\left(8\alpha_1^2 e^{8\alpha_1^2 - 4\beta z}\right)}{8\alpha_1^2} \,, \tag{16}$$

where $w(\cdot)$ is the Lambert function.[1] It is instructive to express this as

$$4\beta z = \frac{8P_{\mathrm{in}}}{P_{\mathrm{sat}}}\left(1 - \frac{P_{\mathrm{el}}(z)}{P_{\mathrm{in}}}\right) - \ln\left(\frac{P_{\mathrm{el}}(z)}{P_{\mathrm{in}}}\right), \tag{17}$$

which reveals a scaling behavior of the particle number (here, $z$) with $\beta$. For low input power ($8P_{\mathrm{in}} \ll P_{\mathrm{sat}}$), the first term can be neglected with respect to the second one, and one recovers the *Beer-Lambert law*

$$P_{\mathrm{el}}(z) \simeq P_{\mathrm{in}} \exp(-4\beta z). \tag{18}$$

For large input powers ($8P_{\mathrm{in}} \gg P_{\mathrm{sat}}$) one finds instead an (initial) nonexponential decay

$$P_{\mathrm{el}}(z) \simeq P_{\mathrm{in}} - \frac{\Gamma z}{2}. \tag{19}$$

In Fig. 2 we illustrate the normalized power of the elastically scattered field $P_{\mathrm{el}}/P_{\mathrm{in}} = \alpha_{N+1}^2/\alpha_1^2$ versus number of atoms (optical depth). We compare results from mean-field theory where $\alpha_{N+1}$ is determined from Eq. (13), with predictions according to the Lambert law (16) and the Beer-Lambert law (18). In the regime of weak coupling and small input power (up to $\beta \lesssim 0.1$ and $P_{\mathrm{in}} \lesssim P_{\mathrm{sat}}$) the Lambert law provides a good approximation for the decay of $P_{\mathrm{el}}$, while the Beer-Lambert law is clearly violated. In Fig. 2, the results of the basic mean-field theory relying on the product state ansatz are compared to and confirmed by those of improved mean-field theories, which will be introduced in Section 4.

---

[1]Here, $w(x)$ denotes the solution of $w\exp(w) = x$ for $x \geq 0$, and fulfills the identities $w(x\exp(x)) = x$ and $\ln w(x) = x - w(x)$.

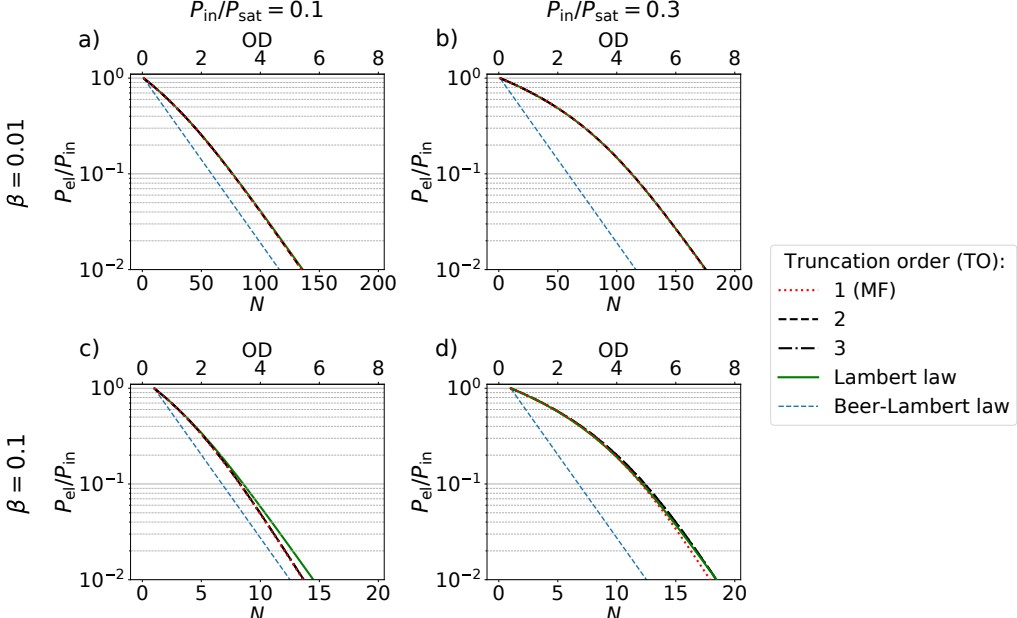

Figure 2: Normalized power of the elastically scattered field $P_{el}/P_{in}$ versus number of atoms (lower $x$-axes) and optical depth (upper $x$-axes). a)-b) correspond to $\beta = 0.01$, c)-d) to $\beta = 0.1$. Columns refer to input powers $P_{in}/P_{sat} = 0.1, 0.3$ from left to right, respectively. Lines show results from mean-field (MF) theory Eq. (15) (dotted red line), Lambert law Eq. (16) (solid green line) and Beer-Lambert law Eq. (18) (dashed blue line). Results from higher order cumulant expansions at truncation order (TO) 2 and 3 are shown as dashed black and dash-dotted black lines, respectively. Mean-field theory gives good results for low $\beta$ and low input power. For $P_{in}/P_{sat} \gtrsim 1/8$ power decay is initially non-exponential, as expressed in Eq. (19).

## 3.2 Squeezing spectra

We next consider the spectra of quadrature fluctuations which is an important quantity that directly reveal quantum features of light. It can be experimentally readily accessed using a balanced homodyne detection scheme, as in [14]. In the following we will first discuss the squeezing spectrum in different truncation order and compare it to experimental data later in Sec 4.2.

In the mean-field approach the product state ansatz implies that in Eq. (7) only the one-particle moments contribute, $\langle\langle \sigma_i^\mu(\tau)\sigma_j^\nu(0)\rangle\rangle = \delta_{ij}\langle\langle \sigma_i^\mu(\tau)\sigma_i^\nu(0)\rangle\rangle$ with $\mu, \nu$ being any $x, y, z$. Therefore, the squeezing spectrum of light after the $j$-th atom is given by the sum of the individual spectra of all $i \leq j$ atoms (cf. (A.3)), where the respective effective driving power is given by (16). That is, in mean-field treatment the problem of computing the squeezing spectrum after $j$ atoms reduces to the problem of resonance fluorescence of $j$ independent atoms, each driven with different power. Squeezing in the resonance fluorescence of single two level atoms has been covered in classic papers by Collet, Walls and Zoller [42,43]. It is shown there that with resonant drive, squeezing only occurs at moderate drive strength $8P_{in}/P_{sat} < 1$, i.e. well below the threshold of $P_{in} \simeq P_{sat}$ at which the Mollow triplet occurs.

Beyond some optical depth, which is dependent on $\beta$ and $P_{in}/P_{sat}$, mean-field theory implies *saturation* in the spectra of the in-phase (in-quadrature) components ($\theta = 0, \pi/2$), as well as in the spectrum of the total inelastically scattered field (8). This is clearly unphysical, since due to the dominant scattering of photons out of the waveguide, the transmitted power must eventually decrease to zero. In Fig. 3 we can clearly observe this effect. There

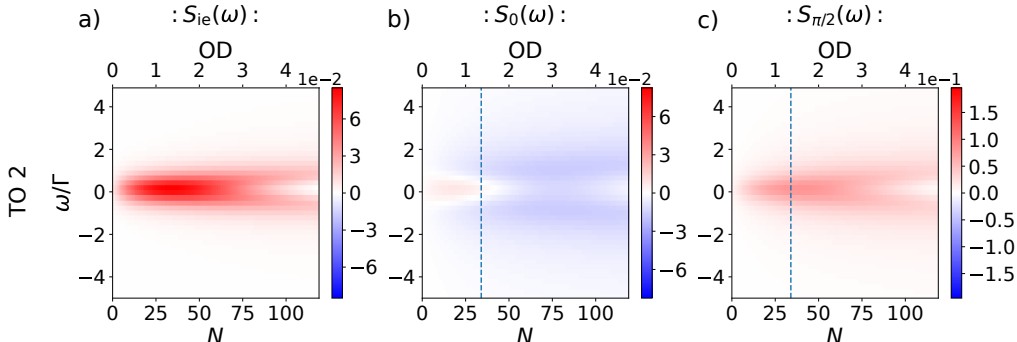

Figure 3: Spectrum of inelastically scattered field $S_{ie}(\omega)$ and of quadrature fluctuations $:S_\theta(\omega):$ for $\beta = 0.01$ and $\frac{P_{in}}{P_{sat}} = 0.1$ calculated in cumulant expansion at TO 2. Normally ordered spectra $:S_\theta(\omega):$ are bounded from below by $-1/4$. Blue regions indicate squeezing. Mean-field theory predicts an unphysical saturation at large optical depth $OD = 4\beta N$ (upper $x$-axes) which is due to the fact that correlations among atoms and re-scattering of photons are not reflected in mean-field approximation. Dashed lines correspond to a point of maximal atomic correlations discussed in Sec. 4 and defined there in Eq. (22).

the spectra of quadrature fluctuations $: S_\theta(\omega) :$ predicted by cumulant expansion at TO 2 are shown for $\beta = 0.01$ and $P_{in}/P_{sat} = 0.1$. The artifact of the mean-field ansatz arises from the implicit assumption that each photon can scatter only once at one of the atoms. The repeated scattering of photons would cause correlations between the atoms that cannot be represented in a mean-field approach. We conclude that mean-field theory, while providing satisfactory results for describing the mean-field amplitude and hence the elastically scattered power, is clearly inadequate for determining squeezing spectra and the inelastically scattered power.

## 4 Higher order cumulant expansions

To incorporate correlations into the description of the system, we use improved mean-field approximations based on a systematic extension of cumulant expansions [34, 36, 37, 39, 44]. The basic mean-field theory described earlier, which neglects all second and higher-order cumulants, corresponds in this framework to a cumulant expansion with truncation order 1 (TO 1). In the following, we will use the cumulant expansions at TO 2, 3, and 4, which account for correlations involving up to two, three and four particles respectively.

In a cumulant expansion at TO 2, all three-body cumulants

$$\langle\!\langle ABC \rangle\!\rangle = \langle ABC \rangle - \langle AB \rangle \langle C \rangle - \langle AC \rangle \langle B \rangle - \langle BC \rangle \langle A \rangle + 2 \langle A \rangle \langle B \rangle \langle C \rangle$$

are discarded, which effectively expresses three-body moments by those of lower order,

$$\langle ABC \rangle \simeq \langle AB \rangle \langle C \rangle + \langle AC \rangle \langle B \rangle + \langle BC \rangle \langle A \rangle - 2 \langle A \rangle \langle B \rangle \langle C \rangle \,. \tag{20}$$

Here, $A$, $B$, and $C$ refer to different atoms. This generalizes the approximation in Eq. (14) from TO 1 to TO 2. By means of Eq. (20), the master equation (1) is approximated by a closed set of differential equations comprised of Eqs. (12) and corresponding equations for two-body-cumulants which we defer to the Appendix A in Eqs. (A.3). This procedure has a systematic extension to higher TO which is discussed in the Appendix. In particular, the generalization of Eqs. (14) and (20) to arbitrary higher order is given in Eq. (A.2) of the Appendix. On this basis

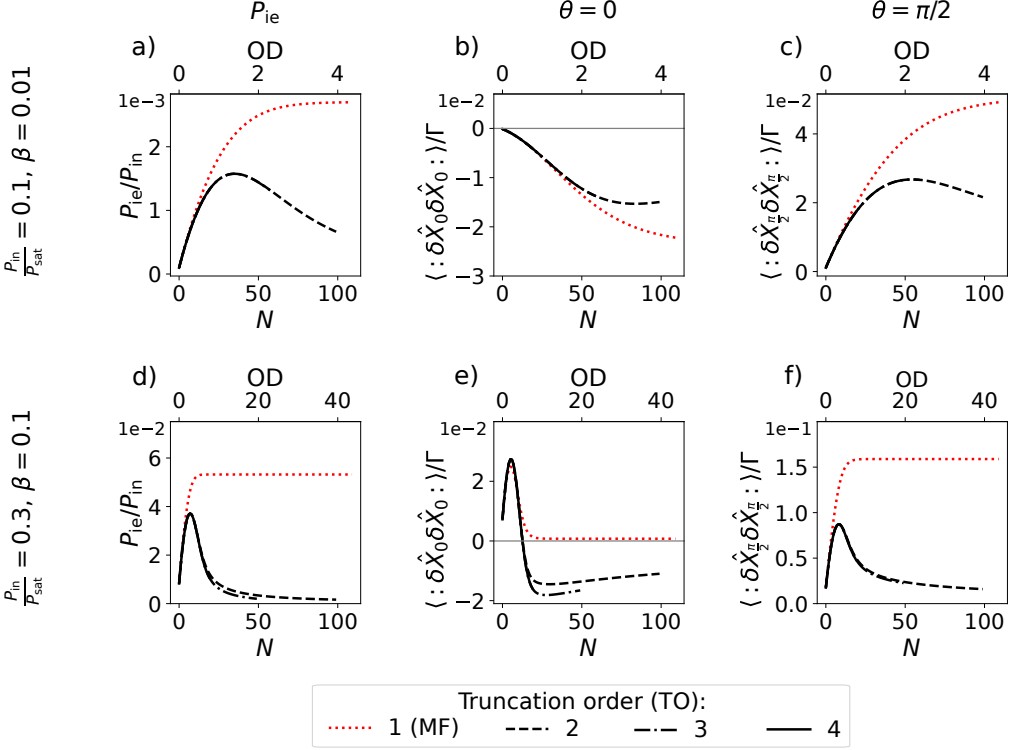

Figure 4: Power of inelastically scattered photons $P_{\text{ie}}$ (left column) and integrated fluctuations $\langle :\delta X_\theta(0)\delta X_\theta(0): \rangle/\Gamma$ for amplitude and phase quadratures (middle and right column), scaled to the atomic line width $\Gamma$, for $\beta = 0.01$, $P_{\text{in}}/P_{\text{sat}} = 0.1$ (top) and $\beta = 0.1$, $P_{\text{in}}/P_{\text{sat}} = 0.3$ (bottom). a) - c) correspond to the frequency integration of the data shown in Fig. 3. Mean-field theory (TO 1, red dotted line) deviates from expansions at TO 2, 3, 4 (black lines) for both sets of parameters. For lower $\beta$ and input power, shown in a) - c), results of truncation order 2 and higher agree. In d) - f) we see that for higher $\beta$ and power even TO 2 and TO 3,4 deviate, indicating that higher order correlations start to play a non-negligible role. Including them raises the computational complexity, which is the reason why the lines for different TO's do not extend equally far.

it is in principle straightforward to derive the corresponding equations for TO 3 and 4 from the master equation (1), but the results are too unwieldy to state explicitly here. The effective dimensionality of the resulting system of equations grows rapidly with increasing TO, which limits the treatment to progressively smaller numbers of particles $N$. In Appendix A we also give a proof for the effective linearity of the system of equations, which is a special feature of cascaded systems at arbitrary order of truncation, and provide further comments and caveats on the method of cumulant expansions.

## 4.1 Transmitted power and squeezing spectra

The results of TO 2 and TO 3 for the power in the elastically scattered field confirm the predictions of mean-field theory and the Lambert law, discussed earlier, for the parameter regime in Fig. 2. However, Fig. 3 shows that the predictions for the squeezing spectra and the power spectrum of the inelastically scattered field differ significantly. In contrast to mean-field theory, a treatment in TO 2 predicts a – physically expected – decay of the spectra, which occurs in particular first at resonant frequencies. The same behavior is observed in the third truncation order, the results of which we show in Fig. 10 in Appendix A. There, a larger range of powers

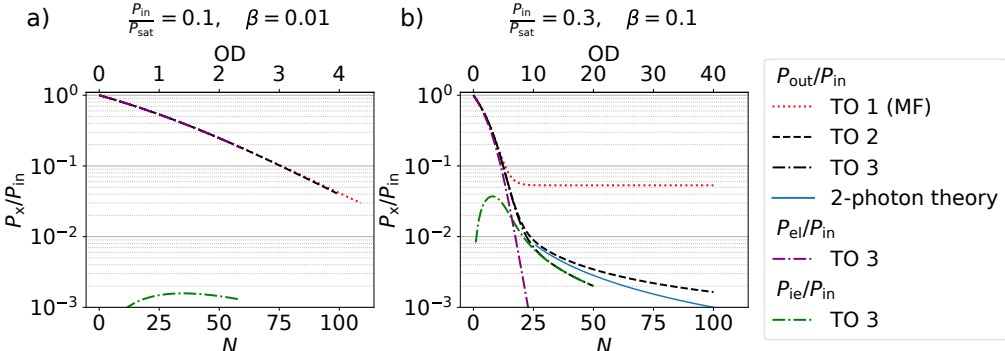

Figure 5: Total output light field $P_{\text{out}} = P_{\text{el}} + P_{\text{ie}}$ for $\beta = 0.01$, $\frac{P_{\text{in}}}{P_{\text{sat}}} = 0.1$ and $\beta = 0.1$, $\frac{P_{\text{in}}}{P_{\text{sat}}} = 0.3$. In a) the elastically scattered part dominates as the fraction of inelastically scattered photons is relatively small. Expressed in other words, this means that the amount of higher order correlations is small and can be truncated without any loss of accuracy. In b) the elastically scattered part dominates in the beginning of the chain as well. But with higher $\beta$ there are more correlations among atoms, i.e. more inelastically scattered light. Here, higher orders of truncation are necessary for higher accuracy. The blue line denotes the high-particle limit of the 2-photon theory of [26].

is considered, covering the transition from squeezing to the Mollow triplet. In the following Section 4.2 we compare the results for squeezing spectra in TO2 to experimental data and find good agreement.

The discrepancy between mean-field theory and higher-order cumulant expansions is further illustrated in Fig. 4, which shows the power in the inelastically transmitted field and its components, the integrated spectra of the quadrature fluctuations defined in Eq. (10). For low $\beta$ and input powers, the higher order truncation results appear to be converged already at TO 2, indicating that it is sufficient to account for two-body correlations. For higher $\beta$ and input powers, it can be seen that the results for TO 2 and TO 3 or TO 4 are in qualitative agreement, but convergence is not yet achieved for TO 2. This testifies the role of atomic correlations, also beyond pair correlations, and the collective nature of the light-atom interaction in the waveguide even at the low coupling strengths considered here. Fig. 4 clearly demonstrates that atomic correlations are essential to obtain a physically meaningful behavior for increasing optical depths and to evade the artificial saturation that occurs in the mean-field approach. Atomic correlations will be explored in more detail in Section 4.3.

It is instructive to combine the results from Figs. 2 and 4 and examine the total output power $P_{\text{out}} = P_{\text{el}} + P_{\text{ie}}$, cf. Fig. 5. For low optical depth, the total transmitted power is dominated by its elastically scattered component. For larger $\beta$ and input powers, a crossover becomes visible at higher optical depths, where the inelastically scattered field becomes dominant. In Fig. 5 we include also the results from the theory developed in [26] where the photon wave function is expanded in the subspace including up to two excitations. This approach is limited to a regime of low (effective) driving power, but appears to be consistent at large optical depths with the asymptotic result of cumulant expansions beyond mean-field theory.

## 4.2 Comparison with experimental data

In the following, we compare the previously obtained results for the squeezing spectrum with experimental data. The waveguide QED platform consists of laser cooled Cesium atoms coupled to a single mode optical nanofiber [14]. The atoms couple weakly to the evanescent field part of the waveguided mode with $\beta = 0.0070(5)$ and yield a total optical depth of $OD \approx 5$.

The atoms are probed with a resonant field that is launched through the fiber with different input powers $P_{in} = 25 - 300\,\text{pW}$. For comparison to experimental data, it is useful to quantify the input power in terms of the saturation parameter $s = \frac{8P_{in}}{P_{sat}}$, which is also consistent with the nomenclature of [14]. The output light is analyzed with a balanced homodyne detection scheme from which we deduce the normally ordered squeezing spectrum $:S_\theta(\omega):$. A more detailed description of the setup and the measurement method can be found in [14]. While the study in [14] was limited to weak excitation regime ($s \ll 1$), the datasets presented here include higher input power but have elsewise been taken under the same conditions. Fig. 6 shows $:S_\theta(\omega):$ for different values of $s$ together with the corresponding theoretical prediction for TO 2. The amplitude ($\theta = 0, \pi$) and the phase quadrature ($\theta = \pi/2, 3\pi/2$) are displayed in blue and orange respectively. For low saturation $s \ll 1$, the amplitude of the squeezing spectrum scales with the input power and is mirror-symmetric with respect to the horizontal axis $:S_\theta(\omega):= 0$, i.e. the noise reduction in one quadrature leads to an increased noise in the conjugate quadrature, c.f. Fig. 6 a) and b). Each spectrum consists of two sidebands which results from the interplay of coherent build-up and absorption. For $s \gtrsim 1$, the atomic transition starts to saturate, which adds additional noise to each quadrature. This behavior is similar to a single emitter [42,45] and breaks the symmetry between the two quadratures, as is apparent from Fig. 6 c) onward. For larger input powers, the experimental conditions are less controlled, resulting in some deviation between theory and experiment. In particular, at higher input powers, photon scattering leads to recoil heating of the atoms, which modifies both $N$

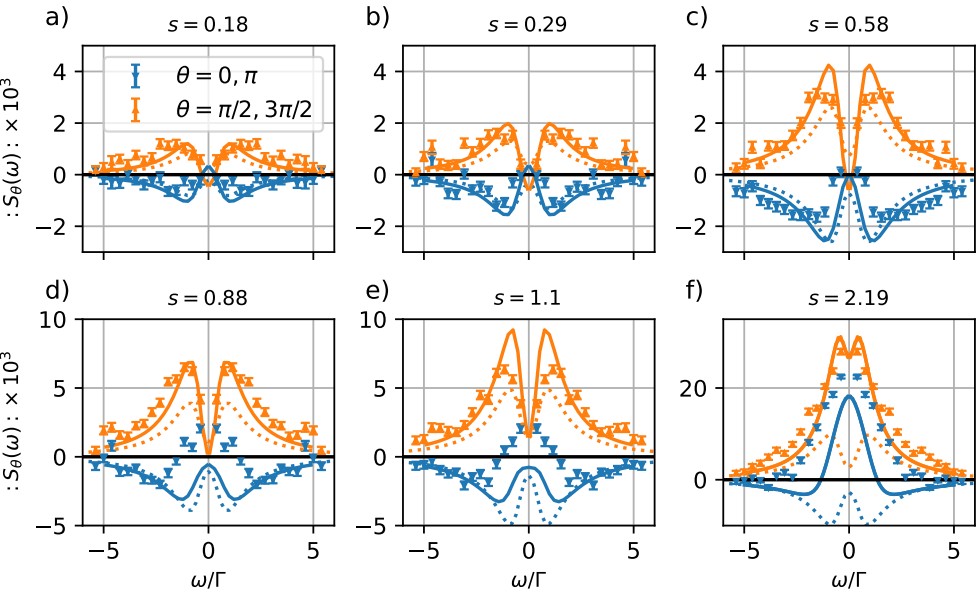

Figure 6: Comparison between experiment and theory for different input saturation parameters $s$ for the amplitude ($\theta = 0, \pi$) and phase quadrature ($\theta = \pi/2, 3\pi/2$) at $OD \approx 5$. The theoretical prediction based on TO2 are shown by the solid lines. At low saturation, as shown in a) and b), the squeezing spectra of the two quadratures are symmetric around $: S_\theta(\omega) := 0$. In this regime the theory predicts a crossing of both quadratures at $\omega \approx 0$, which we attribute to an approximation error due to the high OD. For larger $s$, as shown in c) - f), additional noise appears close to resonance, which breaks the symmetry, and eventually also leads to anti-squeezing for $\theta = 0, \pi$. For comparison we show the prediction in the weak saturation regime $s \ll 1$ [14,26] in dotted lines.

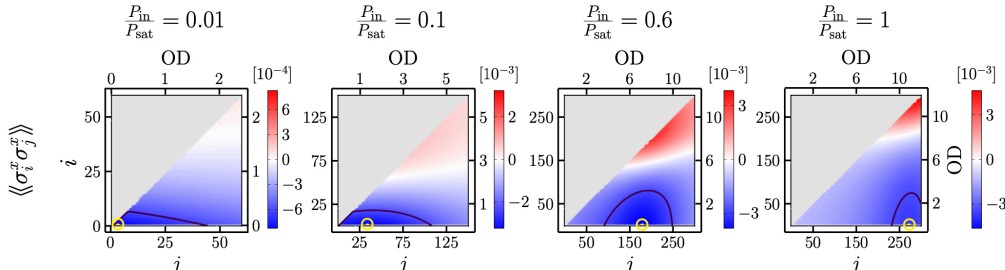

Figure 7: Cumulants $\langle\!\langle \sigma_i^x \sigma_j^x \rangle\!\rangle$ for $\beta = 0.01$ and $\frac{P_{\text{in}}}{P_{\text{sat}}} = 0.01, 0.1, 0.6, 1$ determined in TO 2. The top and right axis show the optical depth $\text{OD} = 4\beta j$. In stationary state, a complex correlation pattern emerges, which also exhibits long range correlations among atoms. The maximal correlation among the first and the $j$–th atom at $j_*$ ($OD_*$) given in Eq. (22) is marked by a yellow circle. As a guide for the eye, the black contour line indicates 70% of the maximal correlation.

and $\beta$, see Appendix A for details. Still, the experimental data exhibits the characteristics predicted by theory: As $s$ increases, additional noise first appears around resonance, breaks the symmetry between the quadratures and eventually, anti-squeezing appears in the amplitude quadrature. We note that in the relevant range, $\beta \ll 1$, the squeezing spectra $S_\theta(\omega)$ do not directly depend on $\beta$, therefore we calculated the spectra at a slightly higher $\beta$ ($= 0.01$), for the same OD, gaining a numerical advantage in terms of a lowered number of atoms.

## 4.3   Atomic correlations

Since correlations among atoms play a crucial role, it is is worth studying them more closely. Fig. 7 shows the pair correlations $\langle\!\langle \sigma_i^x \sigma_j^x \rangle\!\rangle$ in steady state determined in TO 2 for various levels of input power in the regime of weak coupling $\beta = 0.01$. For power levels approaching saturation, a rather complex spin correlation develops along the atoms. Remarkably, even long-range correlations occur, where the pair correlations feature an *extremum* for a certain distance $|i - j|$. For the case of a resonant input field considered here, the equation of motion for $\langle\!\langle \sigma_i^x \sigma_j^x \rangle\!\rangle$ correlations, cf. Eq. (A.3a) in Appendix A, is simple enough to gain some analytical insight regarding this characteristic distance of maximum correlation. In particular, for $i = 1$ one finds the stationary correlation between the first and the $j$–th atom to be well approximated by

$$\langle\!\langle \sigma_1^x \sigma_j^x \rangle\!\rangle = \beta \left( 1 + \langle \sigma_1^z \rangle \right) \langle \sigma_j^z \rangle \prod_{l=2}^{j-1} \left( 1 + \beta \langle \sigma_l^z \rangle \right). \tag{21}$$

Substituting the mean-field solution in (15) for $\langle \sigma_l^z \rangle$, it is possible to approximately determine the index $j_*$ where these correlations become extremal. One finds that this is the case at an optical depth

$$OD_* = 4\beta j_* \simeq \ln\left( \frac{24 P_{\text{in}}}{P_{\text{sat}}} \right) + \frac{8 P_{\text{in}}}{P_{\text{sat}}} + 2\beta - \frac{1}{3}. \tag{22}$$

We note that this formula holds in the limit of small $\beta$ and does not cover the limit $P_{\text{in}} \to 0$ where correlations decay monotonically. A comparison of this formula to numerical results is given in Fig. 7 and shows good agreement. We also mark the optical depth $OD_*$ in the squeezing spectra shown in Figs. 3 and 10. We observe that it correlates with the cross over of $:S_0(0):$ from antisqueezing to squeezing. It would be highly interesting to have a similar

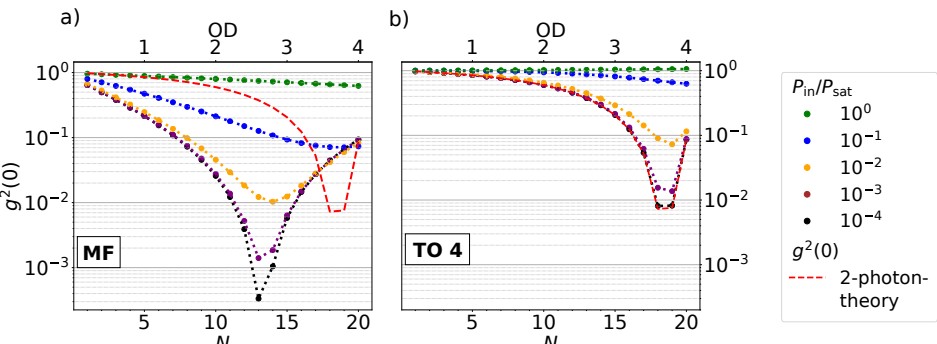

Figure 8: $g^2(0)$ correlation function for $\beta = 0.05$, different input powers and in a) truncation orders 1 (MF) and in b) TO 4.

characterization of the maximal correlations of $\langle\langle \sigma_i^y \sigma_j^y \rangle\rangle$, since these determine – for resonant driving – at which optical depths maximum squeezing occurs in $:S_0(0):$. Unfortunately, the equations of motion for this case are more complex, cf. Eqs. (A.3), and cannot be solved in the same way.

## 4.4 Second-order coherence

Finally, we extend our treatment further to a cumulant expansion at TO 4. This enables us to discuss the second-order coherence function $g^2(0)$ and the antibunching of transmitted photons. This was recently investigated and experimentally demonstrated in [13]. The experimental results were compared to the two-photon theory of [26] showing good agreement after taking into account an uncertainty in OD. The experiments were conducted for low coupling strength $\beta = 0.0083 \pm 0.0003$ and low input power $s = 0.02$ ($P_{in}/P_{sat} = 0.0025$).

One can expect to see a rising discrepancy between the experimental results and 2-photon theory for higher input power. Our work is complementary in the sense that we can study the system at higher powers. However, the scaling of the effective dimensionality restricts our treatment to moderate particle numbers. This ensues that in the regime of low coupling strength ($\beta \leq 0.01$) it becomes unfeasible to investigate the optical depth at which antibunching is maximal. Nevertheless, for slightly larger coupling strengths $\beta \geq 0.05$ we are able to treat optical depths of interest showing good agreement between our approach in low-power (black line at TO 4) and the 2-photon theory (red line), cf. Fig. 8. $g^2(0)$ is given in Eq. (11) and expressed in terms of atomic moments in Eq. (A.5).

Since $g^2(0)$ depends on correlations up to fourth order, low-order truncations can be expected not to yield reliable results. In Fig. 8 we compare the results of mean-field theory and cumulant expansions at higher order for $\beta = 0.05$ and various levels of input power. Surprisingly, a mean-field approximation does give qualitatively similar results as higher order truncations. However, it is quantitatively wrong in the sense that it predicts too strong antibunching at too small optical depth. Here, as in computing other observables, mean-field theory means effectively a factorization of higher-order moments into products of first-order moments, c.f. (A.5). Results at TO 2 turned out to be nonphysical (predicting negative values for $g^2(0)$), and are therefore not shown in Fig 8. We attribute this unphysicality to the fact that one needs to apply a *nested* cumulant expansion in order to compute the 4-body moments in (A.5) at TO 2. Higher order expansions at TO 3 and 4 do not require nested expansions, and give physical results. They show good agreement among each other and, at low powers, with the predictions of [26].

Thus, in order to get a quantitatively correct description, the inclusion of higher order correlations is essential. As we saw in Fig. 4 and 5, correlations account for the initial collectively

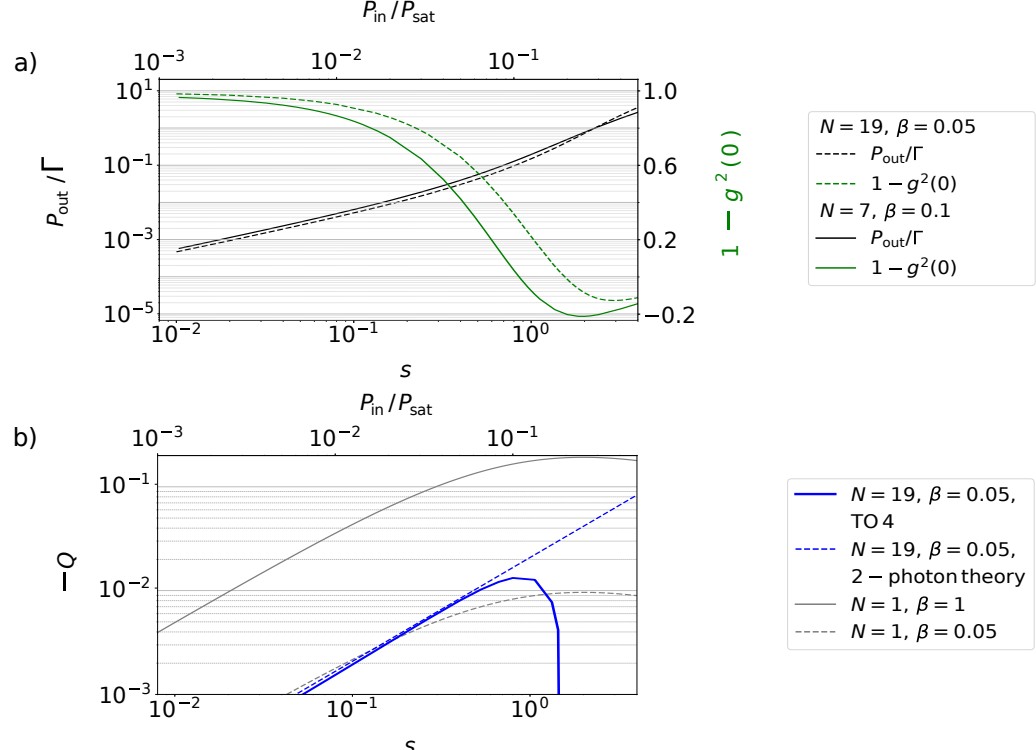

Figure 9: a) shows the output power $P_{\text{out}}/\Gamma$ in dependence of the input power at maximal antibunching, i.e. $N = 19$ for $\beta = 0.05$ and $N = 7$ for $\beta = 0.1$, illustrating the trade-off between large anti-bunching and large output-power. b) compares the performance of single-photon sources based on single emitters (grey), with $\beta = 0.05$ (dashed), $\beta = 1$ (solid) and collective sources (blue) for $\beta = 0.05$ calculated in TO4 (solid) and using the 2-photon theory (dashed).

enhanced build up of the inelastically scattered part $P_{\text{ie}}$ and its subsequent loss. Antibunching can be understood as a delicate interference between the elastically and inelastically scattered components. Therefore, including correlations alters the prediction of occurrence and magnitude of antibunching strongly.

Fig. 9 shows the output power $P_{\text{out}}/\Gamma$ (black line) in dependence of the input power at maximal antibunching, i.e. $N = 19$ for $\beta = 0.05$ and $N = 7$ for $\beta = 0.1$. In the same plot, the green line shows $1 - g^2(0)$ illustrating the amount of antibunching in the output field at different input powers. Evidently, there is a trade-off between the level of antibunching and output power, which is important to grasp if the system is considered as a single-photon source [40]. Following up on this idea, it is worth comparing the performance of such a collective single photon source with that of a reference source based on a continuously driven single atom with a linewidth-limited transition whose emitted photons are collected into a given optical mode. In principle, one has to compare two quantities: indistinguishability of the photons and the achievable photon output rate, or brightness. Absent inhomogeneous spectral broadening, the photon current generated with the collective scheme has a similar spectral width and yields a similar temporal shape of $g^2(\tau)$ as the fluorescence of a single atom. When transforming the photon current into a train of pulses containing at most one photon, both approaches thus yield the same performance in terms of photon indistinguishability.

In order to quantify the brightness of both types of sources, we require a quantity that depends on the photon output rate $P_{\text{out}}$ and temporal width $\tau$ of the anti-bunching dip. The latter defines the timescale over which one can be sure that, given a photon detection event,

no second photon detection will occur. Therefore, we define the quantity

$$Q = P_{\text{out}} \cdot \tau \left( g^{(2)}(0) - 1 \right), \tag{23}$$

where $\tau$ defines the full width of the anti-bunching dip where it reaches 85% of its maximum depth. In this region, we approximate $g^{(2)}(t)$ to be constant. For a single atom at low driving one can show that this definition of $\tau$ yields $\tau = 1/\Gamma$. The quantity $Q$ is equivalent to the continuous wave version of the Mandel Q parameter [46] which quantifies the deviation of the photon statistics of the light from a Poissonian distribution in the time interval $2\tau$. For a single-atom-based source with photon collection efficiency $\beta$, we obtain the analytical expression $Q = -\beta s / 2(1 + s)^{3/2}$ with a minimum value of $Q_{\text{min}} = -0.19\beta$. For the collective source, the formalism presented in this manuscript opens up the path to an investigation of the rate–quality trade-off also in the regime of strong driving. As the performance of the collective source is in first approximation independent of $\beta$, we calculate the expected Q-parameter as a function of the input power for the experimental parameters underlying Fig. 8, see section A. The result is shown in the second panel of Fig. 9. The minimum value of $Q$ for this type of source is $Q_{\text{min}} = -0.013$. This is about 6.5% of that of a perfect single photon source, i.e. a single emitter-based source with unit collection efficiency. In other words, this means that at the optimal working point the performance of a collective single photon source is equivalent to that of a single emitter based source with $\beta = 6.5\%$. This shows that such a collective single photon source outperforms single quantum emitter-based photon sources in situations where $\beta$ factors larger than 0.065 cannot be realized.

# 5 Conclusions and outlook

We employed an improved mean-field theory based on a higher order cumulant expansion to determine the stationary state of a strongly-driven, weakly-coupled, chiral chain of atoms. We inferred the power of the transmitted light, its elastic and inelastic component, as well as squeezing spectra and the degree of second-order coherence. Our treatment evidences the important role of atomic correlations of growing order for larger input powers. Thanks to the linearity of the effective equations of motion, we are able to compare different order of cumulant expansions, and in this sense investigate systematically the deviations from a classical, mean-field description. We find that the system develops intriguing long-range correlations in steady state. Our theoretical predictions regarding squeezing spectra agree well with experimental results, even for large powers that could not be captured in previous descriptions.

Our approach can be extended in various directions. Firstly, the assumption of resonant drive can be easily dropped, without changing our treatment conceptually. Secondly, the trade-off between anti-bunching and photon flux can be investigated more systematically. In order to do so for lower coupling, our approach need to be made more efficient in terms of the scaling with particle number, at least at TO 3. This could be done by restricting the descriptions to those three-particle correlations which are making a relevant contribution to $g^2(0)$. Thirdly, while we focused here on a perfectly unidirectional system, it would be interesting to consider also systems of mixed chirality. This would, however, come at the cost of an unavoidable nonlinearity in the equations to be solved.

# Acknowledgement

This work was funded by the Deutsche Forschungsgemeinschaft (DFG, German Research Foundation) under SFB 1227 'DQ-mat' project A06 - 274200144 and Germany's Excellence Strategy - EXC-2123 QuantumFrontiers - 390837967, the Alexander von Humboldt Foundation in the framework of the Alexander von Humboldt Professorship endowed by the Federal Ministry of Education and Research, as well as the European Commission under the project DAALI (No.899275). M. C. and M. S. acknowledge support by the European Commission (Marie Skłodowska-Curie IF Grant No. 101029304 and IF Grant No. 896957).

# A   Appendix

## General idea of cumulant expansions

We first review the general idea behind a cumulant expansion, for which we also refer to [36] and references in there. For $N$ particles, an $(\ell+1)$-body moment is given by $\langle \otimes_{m=1}^{\ell+1} \sigma_{j_m}^{\beta_m} \rangle$ where we take $\ell+1 \leq N$, $\beta_m \in \{x,y,z\}$, $j_m \in [1,N]$ and $j_m \neq j_n$ for $m \neq n$. The corresponding $(\ell+1)$-body cumulant is defined by [47]

$$\langle\!\langle \bigotimes_{m=1}^{\ell+1} \sigma_{j_m}^{\beta_m} \rangle\!\rangle = \sum_{P \in P_{\ell+1}} f(|P|) \prod_{M \in P} \langle \bigotimes_{n \in M} \sigma_{j_n}^{\beta_n} \rangle, \tag{A.1}$$

where $P_{\ell+1}$ denotes the set of all partitions of the interval $[1,\ell+1]$, and $f(n) = (-1)^{n-1}(n-1)!$. Note that one of the elements in $P_{\ell+1}$ is the trivial partition given by $\{[1,\ell+1]\}$. This is the only partition with $|P| = 1$, and contributes the $(\ell+1)$-body moment on the right hand side of Eq. (A.1).

In an expansion at truncation order (TO) $\ell$, cumulants of order $\ell+1$ are set to zero. This is equivalent to setting

$$\langle \bigotimes_{m=1}^{\ell+1} \sigma_{j_m}^{\beta_m} \rangle = - \sum_{\substack{P \in P_{\ell+1} \\ |P|>1}} f(|p|) \prod_{M \in P} \langle \bigotimes_{n \in M} \sigma_{j_n}^{\beta_n} \rangle, \tag{A.2}$$

which effectively replaces correlations of order $\ell+1$ by a nonlinear function of correlations of lower order. In this way, the master equation is approximated by a system of differential equations of lower dimension (depending on the TO), which is closed but generally nonlinear. In the next section we will explain that this is not the case for cascaded systems.

Before that, we comment on some well-known problems with cumulant expansions, and explain how we are dealing with these issues in this work. As was explained Sec. 3, a cumulant expansion at TO 1 corresponds to a product state ansatz for the density matrix, which is a physically meaningful state by construction. Cumulant expansions at higher order have no such analog on the level of the density matrix. Thus, the correlations determined in this way do not necessarily correspond to a physical state. Whether or not a given set $\mathcal{C}_\ell^n$ of correlations up to a certain order $\ell$ among $n$ particles can be due to a density operator $\rho_n$ corresponds to the quantum marginal problem [48–50], which is NP-hard and even QMA-complete.

In the present context the unphysicality of the set of correlations $\mathcal{C}_\ell^n$ can give rise to unphysicalities in the properties of the transmitted field. For examples, it can give rise to negative expectation values of positive quantities such as transmitted power, or to violations of Heisenberg uncertainty relations for quadrature fluctuations, which reads $(:S_0(\omega):)(:S_{\pi/2}(\omega):) \geq \frac{1}{16}$. In our truncations up to TO 4 such violations do occur for large $\beta$ and input powers, that is, in

regimes where correlations of even higher order become important. Comparing results from different expansions confirms that unphysicalities become less severe or disappear at higher TO. In all cases shown and discussed here, we have ensured that the power spectrum is positive everywhere and that the Heisenberg uncertainty of the squeezing spectra is satisfied.

### Equations of motion in second order cumulant expansion

For example, in TO 2, the master equation (1) implies the equations of motion for second order cumulants

$$
\frac{1}{\Gamma}\frac{d}{dt}\langle\!\langle\sigma_i^x\sigma_j^x\rangle\!\rangle = -\langle\!\langle\sigma_i^x\sigma_j^x\rangle\!\rangle + \beta\langle\!\langle\sigma_i^z\sigma_j^z\rangle\!\rangle + \beta\langle\sigma_i^z\rangle\langle\sigma_j^z\rangle + \beta\sum_{l=1}^{i-1}\langle\!\langle\sigma_l^x\sigma_j^x\rangle\!\rangle\langle\sigma_i^z\rangle
$$
$$
+ \beta\sum_{l=1}^{j-1}\langle\!\langle\sigma_l^x\sigma_i^x\rangle\!\rangle\langle\sigma_j^z\rangle\,, \tag{A.3a}
$$

$$
\frac{1}{\Gamma}\frac{d}{dt}\langle\!\langle\sigma_i^y\sigma_j^y\rangle\!\rangle = -\langle\!\langle\sigma_i^y\sigma_j^y\rangle\!\rangle - 2\alpha_j\langle\!\langle\sigma_i^y\sigma_j^z\rangle\!\rangle - 2\alpha_i\langle\!\langle\sigma_i^z\sigma_j^y\rangle\!\rangle + \beta\langle\!\langle\sigma_i^z\sigma_j^z\rangle\!\rangle + \beta\langle\sigma_i^z\rangle\langle\sigma_j^z\rangle
$$
$$
+ \beta\sum_{l=1}^{i-1}\langle\!\langle\sigma_l^y\sigma_j^y\rangle\!\rangle\langle\sigma_i^z\rangle + \beta\sum_{l=1}^{j-1}\langle\!\langle\sigma_l^y\sigma_i^y\rangle\!\rangle\langle\sigma_j^z\rangle\,, \tag{A.3b}
$$

$$
\frac{1}{\Gamma}\frac{d}{dt}\langle\!\langle\sigma_i^y\sigma_j^z\rangle\!\rangle = -\frac{3}{2}\langle\!\langle\sigma_i^y\sigma_j^z\rangle\!\rangle - 2\alpha_i\langle\!\langle\sigma_i^z\sigma_j^z\rangle\!\rangle + 2\alpha_j\langle\!\langle\sigma_i^y\sigma_j^y\rangle\!\rangle - \beta\langle\!\langle\sigma_i^z\sigma_j^y\rangle\!\rangle - \beta\langle\sigma_i^z\rangle\langle\sigma_j^y\rangle
$$
$$
+ \beta\sum_{l=1}^{i-1}\langle\!\langle\sigma_l^y\sigma_j^z\rangle\!\rangle\langle\sigma_i^z\rangle - \beta\sum_{l=1}^{j-1}\langle\!\langle\sigma_l^y\sigma_i^y\rangle\!\rangle\langle\sigma_j^y\rangle\,, \tag{A.3c}
$$

$$
\frac{1}{\Gamma}\frac{d}{dt}\langle\!\langle\sigma_i^z\sigma_j^z\rangle\!\rangle = -2\langle\!\langle\sigma_i^z\sigma_j^z\rangle\!\rangle + 2\alpha_i\langle\!\langle\sigma_i^y\sigma_j^z\rangle\!\rangle + 2\alpha_j\langle\!\langle\sigma_i^z\sigma_j^y\rangle\!\rangle + \beta\langle\!\langle\sigma_i^x\sigma_j^x\rangle\!\rangle + \beta\langle\sigma_i^x\rangle\langle\sigma_j^x\rangle
$$
$$
+ \beta\langle\!\langle\sigma_i^y\sigma_j^y\rangle\!\rangle + \beta\langle\sigma_i^y\rangle\langle\sigma_j^y\rangle - \beta\sum_{l=1}^{j-1}\langle\!\langle\sigma_l^y\sigma_i^z\rangle\!\rangle\langle\sigma_j^y\rangle - \beta\sum_{l=1}^{i-1}\langle\!\langle\sigma_l^y\sigma_j^z\rangle\!\rangle\langle\sigma_i^y\rangle\,, 
$$
$$
\tag{A.3d}
$$

for $i \neq j$. Here, as in (12), we left quantities of the form $\langle\!\langle\sigma_i^x\sigma_j^{y,z}\rangle\!\rangle$ out, since numerical evidence shows that these vanish. These equations complement and close those in Eq. (12). We call attention to the nonlinearity on the right hand side which arises from the cumulant expansion. To gain insight into the correlation structure $\langle\!\langle\sigma_1^x\sigma_j^x\rangle\!\rangle$ and derive Eq. (21) we set $\beta\langle\!\langle\sigma_i^z\sigma_j^z\rangle\!\rangle$ to zero. This approximation is justified by the fact that $\langle\!\langle\sigma_i^z\sigma_j^z\rangle\!\rangle$ cumulants are by themself small (numerical evidence) and multiplied by $\beta$ this term gets neglectable.

### Linearity of cumulant expansions for cascaded systems

The cascaded nature of the dynamics implies that the state $\rho_n$ of the *first n* atoms evolves autonomously and independently of the $N - n$ atoms to the right. This can be seen by taking the partial trace with respect to these $N - n$ atoms in the master equation (1), which gives, for all $n \leq N$,

$$
\frac{1}{\Gamma}\frac{d\rho_n}{dt} = L_n\rho_n\,. \tag{A.4}
$$

An important consequence of this property, which holds generally for any cascaded system, is the following: The cumulant expansion at any order of a cascaded system yields a nonlinear system of differential equations whose structure corresponds to a hierarchy of nested systems of actually *linear* differential equations. This feature prevents certain numerical difficulties

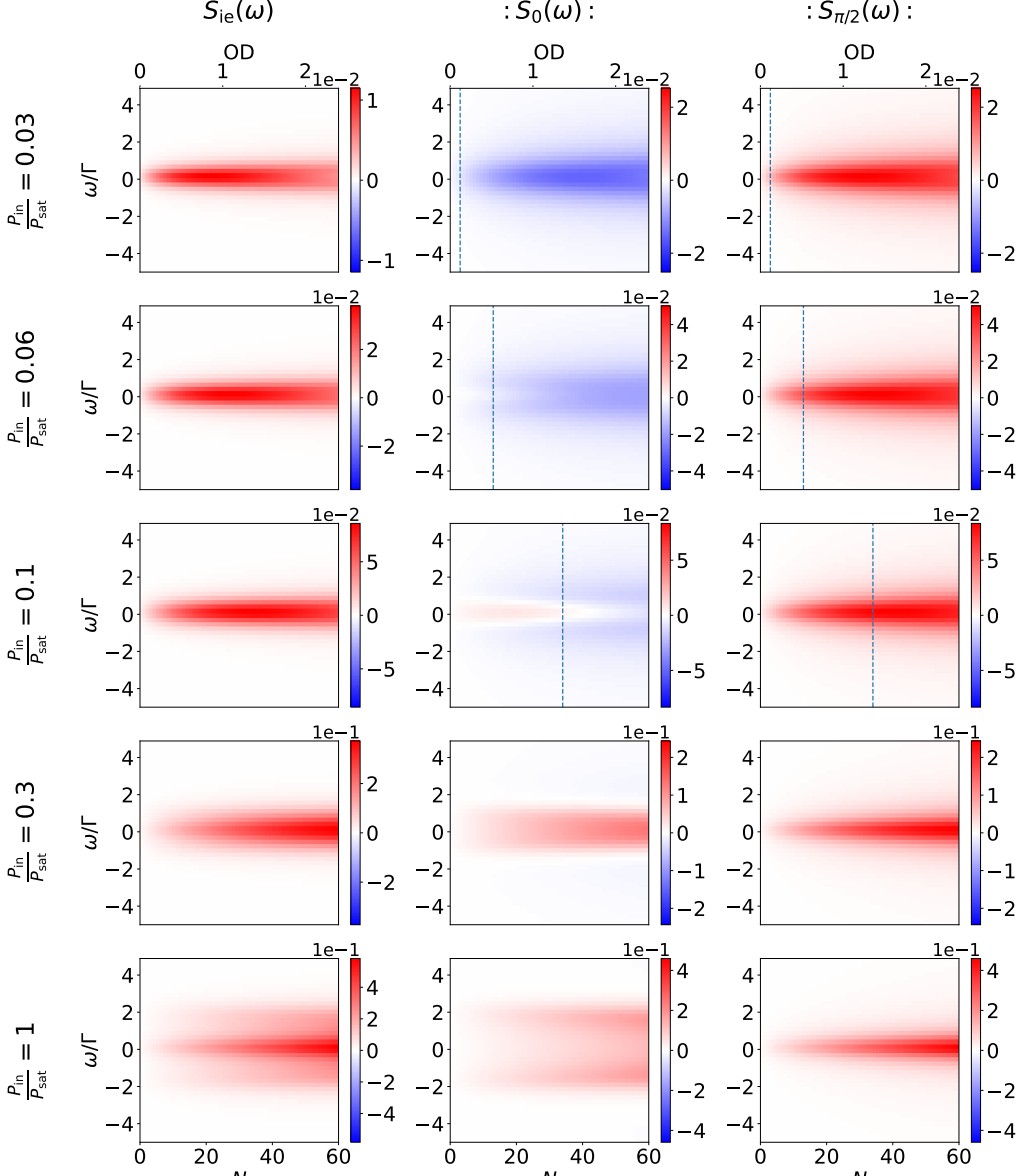

Figure 10: $S_{\text{ie}}(\omega)$, $:S_0(\omega):$, $:S_{\pi/2}(\omega):$ for $\beta = 0.01$ and $P_{\text{in}}/P_{\text{sat}} = 0.03, 0.06, 0.1,$ 0.3, 1 at TO 3. The dashed line is computed via equation (22). As observed earlier in Fig. 6, in the low power regime the spectra are symmetric. Approaching a regime where $P_{\text{in}} \simeq P_{\text{sat}}$ we see the transition to and, eventually, in the last row, the manifestation of the Mollow triplet.

in solving the truncated differential equation systems associated with the normally occurring nonlinearity, which becomes pertinent especially for higher TOs.

In order to show this, we denote by $\mathcal{C}_\ell^n$ the set of correlations of the type (A.1) involving up to (and including) $\ell$ particles among the leftmost $n$ atoms, assuming $\ell \leq n$. Thus, $\mathcal{C}_\ell^n$ contains at most $\ell$-body moments and it is a subset of $\mathcal{C}_\ell^{n+1}$. $\mathcal{C}_\ell^{n+1}$ contains additionally all correlations up to order $\ell$ involving the $(n+1)$-th particle.

Our argument proceeds inductively: The correlations $\mathcal{C}_\ell^\ell$ up to order $\ell$ among the first $\ell$ atoms follow without approximation from Eq. (A.4) for $n = \ell$. Thus, $\mathcal{C}_\ell^\ell$ can be determined by solving linear equations. Now suppose that in a cumulant expansion of order $\ell$ the correlations $\mathcal{C}_\ell^n$ can be determined by solving linear equations, which, as we have just seen, holds for

$n = \ell$. To determine the correlations in $\mathcal{C}_\ell^{n+1}$, we additionally need all correlations involving the $(n+1)$-th particle. These satisfy linear equations of motion involving all correlations in $\mathcal{C}_{\ell+1}^{n+1}$ involving up to $\ell+1$ particles. In a cumulant expansion at order $\ell$, $(\ell+1)$-particle correlations are replaced by Eq. (A.2). In the right hand side of Eq. (A.2) each term in the sum contains exactly *one* factor involving the $(n+1)$-th particle. All other factors are elements of $\mathcal{C}_\ell^n$, which are known. This implies that the correlations involving the $(n+1)$-th particle follow from linear equations, and so does $\mathcal{C}_\ell^{n+1}$.

### Degree of second order coherence

Inserting the input-output relation (2) in the definition of $g^2(0)$ in Eq. (11) one arrives at

$$
\begin{aligned}
g^2(0) = \frac{P_{\text{in}}^2}{P_{\text{out}}^2} \Bigg( & 1 - 2\frac{\Gamma}{\sqrt{P_{\text{in}}/P_{\text{sat}}}} \sum_j \langle \sigma_y^j \rangle + 4\frac{\Gamma^2}{P_{\text{in}}/P_{\text{sat}}} \sum_j \langle \sigma_{ee}^j \rangle + \frac{\Gamma^2}{2 P_{\text{in}}/P_{\text{sat}}} \sum_{\substack{ij \\ i \neq j}} \left( \langle \sigma_x^i \sigma_x^j \rangle + 3\langle \sigma_y^i \sigma_y^j \rangle \right) \\
& - 4\frac{\Gamma^3}{2(P_{\text{in}}/P_{\text{sat}})^{3/2}} \sum_{\substack{ijk \\ i \neq j \neq k}} \left( \langle \sigma_y^i \sigma_x^j \sigma_x^k \rangle + \langle \sigma_y^i \sigma_y^j \sigma_y^k \rangle \right) \\
& - 2\frac{\Gamma^3}{(P_{\text{in}}/P_{\text{sat}})^{3/2}} \sum_{\substack{ij \\ i \neq j}} \langle \sigma_{ee}^i \sigma_y^j \rangle + \frac{2\Gamma^4}{P_{\text{in}}^2/P_{\text{sat}}^2} \sum_{\substack{ij \\ i \neq j}} \langle \sigma_{ee}^i \sigma_{ee}^j \rangle \\
& + \frac{\Gamma^4}{2 P_{\text{in}}^2/P_{\text{sat}}^2} \sum_{\substack{ijk \\ i \neq j \neq k}} \left( \langle \sigma_x^i \sigma_x^j \rangle + \langle \sigma_y^i \sigma_y^j \rangle + \langle \sigma_z^i \sigma_x^j \sigma_x^k \rangle + \langle \sigma_z^i \sigma_y^j \sigma_y^k \rangle \right) \\
& + \frac{\Gamma^4}{16 P_{\text{in}}^2/P_{\text{sat}}^2} \sum_{\substack{ijkl \\ i \neq j \neq k \neq l}} \left( \langle \sigma_x^i \sigma_x^j \sigma_x^k \sigma_x^l \rangle + 2\langle \sigma_x^i \sigma_y^j \sigma_x^k \sigma_y^l \rangle + \langle \sigma_y^i \sigma_y^j \sigma_y^k \sigma_y^l \rangle \right) \Bigg),
\end{aligned}
\tag{A.5}
$$

where we used $\sigma_{ee} = (1 + \sigma_z)/2$ to denote the projector on the excited state. In mean-field approximation, every moment of order higher than one factorizes into products of first order moments, and the expression for $g^2(0)$ simplifies considerably due to $\langle \sigma_x^i \rangle = 0$ for resonant drive. In treatments at truncation order $\ell$, all correlations involving more than $\ell$-particles are approximated by those of lower order by means of Eq. (A.2).

### Squeezing spectra in third order cumulant expansion

Here, we complement Fig. 3 and show results for squeezing spectra in cumulant expansion at TO 3, which evidences good convergence at TO 2. We also extend the analysis to a larger set of input powers approaching saturation, where the Mollow triplet emerges in the spectrum of the inelastically scattered field. The results are shown in Fig. 10.

### Experimental details

**The experimental platform**  The experimental platform consists of a nanofiber-based optical interface for laser-cooled Cesium (Cs) atoms. The optical nanofiber with 400 nm diameter is realized as the waist of a tapered optical fiber. The atoms are trapped using the evanescent field surrounding the nanofiber where two 1D arrays are formed along the nanofiber through a combination of red- and blue-detuned nanofiber-guided light fields [51–54]. The atoms are located at a distance of $\sim 250$ nm from the nanofiber surface. Each trapping site contains at most one atom and the average filling fraction is about 10 %. A nanofiber-guided probe field of power $P_{\text{in}}$, which is resonant with the Cesium D2-line ($F = 4 \to F' = 5$) transition, is launched through the tapered optical fiber and interfaces the atoms via the evanescent field

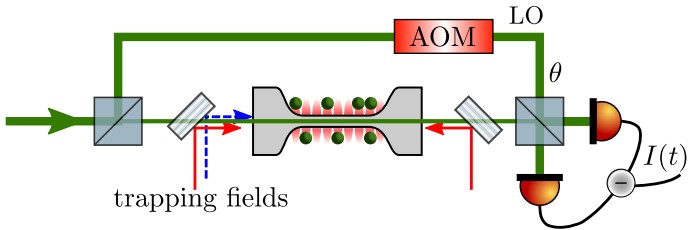

Figure 11: Experimental setup to measure squeezing of light upon propagating through an array of nanofiber-trapped atoms. The probe light interacts via the evanescent field of the nanofiber. After the interaction, the probe light interferes with a local oscillator (LO) with a relative phase of $\theta$ on a 50:50 beam splitter and the resulting light is analyzed on balanced photo-detectors. The atoms are trapped in the evanescent field surrounding the nanofiber-waist of a tapered optical fiber by a combination of a red-detuned standing-wave light field at a free-space wavelength of 935 nm (solid red line) and blue-detuned running-wave light field (dashed blue line) at 685 nm. Figure adapted from [14].

of the nanofiber mode, see Fig. 11. The coupling of individual atoms to the nanofiber mode is characterized by the coupling constant $\beta = \Gamma_{\mathrm{wg}}/\Gamma_{\mathrm{tot}} = 0.0070(5)$, where $\Gamma_{\mathrm{wg}}$ is the spontaneous emission rate into the waveguide and $\Gamma_{\mathrm{tot}} = 2\pi \times 5.2\,\mathrm{MHz}$ is the total emission rate. We analyze the transmitted light via a balanced homodyne detection scheme: First, the trapping light fields are suppressed in the output by means of optical filters and mixed with a local oscillator (LO) field on a 50:50 beam splitter. The powers at the two outputs are recorded on balanced photo-detectors, from which we deduce the amplified differential current between both photodiodes, $I(t)$. From this differential current, we deduce the squeezing spectrum, $S_\theta(\omega)$, and normalized it to the spectrum of a coherent state. In the final step, we deduce the normally ordered squeezing spectrum, $:S_\theta(\omega): = \int \langle :\hat{X}_\theta(0)\hat{X}_\theta(\tau): \rangle e^{i\omega\tau}\mathrm{d}\tau$, from $S_\theta(\omega)$. The optical depth, $OD$, and $\beta$ are both determined in a separate transmission measurement. More details can be found in [14].

**Heating and probing time**   We probe the atoms with input powers ranging from 20-300 pW during $10 - 100\,\mu s$ and repeat the experiment 10 000 - 100 000 times. For larger input power, heating of the atoms during probing becomes important. To avoid a too large temperature, we decrease the probing time for larger input power: Up to $s = 0.6$ we probe for $100\,\mu s$ and then gradually decrease the probing time to keep the OD approximately constant. For the largest saturation parameter $s = 2.19$, we probe for $10\,\mu s$ and the OD changes by up to 20%.

Even with reduced probing times, we expect that heating is the main source for the discrepancy between theory and experiment at high input power since it affects both $\beta$ and $N$. First, the atoms are confined in anharmonic traps, such that the average coupling constant $\beta$ decreases for atoms with larger energy. Second, atoms can be lost from the trap, which has a finite depth of about $\simeq 100\,\mu K$. Modeling how this modifies the squeezing spectrum is beyond the scope of this work.

**Squeezing angle**   On resonance ($\Delta = 0$), the interesting squeezing angles are at $\theta = 0$ and $\theta = \pi/2$ for which the largest squeezing and anti-squeezing occurs. In order to increase the signal to noise in $S_\theta(\omega)$, we use the $\pi$-periodicity of $S_\theta(\omega)$ and average the data over $\theta = 0$ and $\pi$ as well as $\theta = \pi/2$ and $3\pi/2$ respectively. For each value of $\theta$, we average over a range of $\pm 18°$ [14].

**The saturation parameter $s$ vs. $P_{in}/P_{sat}$**   We point out that we characterize the saturation of the emitters by two quantities, $s$ and $P_{in}/P_{sat}$, depending on the context. Both quantities are linked by $s = 8P_{in}/P_{sat}$. The saturation parameter $s$ is more convenient when referring to experimental data. For $s = 1$, an emitter is subject to a light intensity $I_{sat}$ and scatters $\Gamma/4$ photons [55]. The saturation power $P_{sat} = \frac{\Gamma}{\beta}$ is more suitable for writing formulas, such as equations of motion etc., where it simplifies notations [26]. We remind the reader that we scale powers to photon flux by $\hbar\omega_0$.

### Quantifying the performance of source of a antibunched light

For practical implementations of single photon sources based on a stream of antibunched light, it is crucial to provide a physical parameter that quantifies the latter's performance. This parameter should be linear in the output photon flux $P_{out}$. Furthermore, it should quantify how much the photon statistics is different from a Poissonian distribution, i.e. the larger the temporal width of the antibunching dip of $g^{(2)}(t)$, the longer the output fields remains non-classical after the detection of a single photon and the higher the quality of the source. Here, we chose the Mandel-Q factor that quantifies the deviation of the photon statistics of a light field from a Poissonian distribution [46]

$$Q = P_{out} \int_{-\tau}^{\tau} dt \left(1 - \frac{|t|}{\tau}\right)\left(g^{(2)}(t) - 1\right),\tag{A.6}$$

where $Q < 0$ indicates a sub-Poissonian photon statistics. In the following, we consider a sufficiently short time interval after the detection of a photon, so that $g^{(2)}(t) \approx$ const. For this, we define $\tau$ as the 85% width of the anti-bunching dip. In this approximation one obtains

$$Q \approx P_{out}\tau\left(g^{(2)}(0) - 1\right).\tag{A.7}$$

For the fluorescence of a single atom, $\tau$ is given by $\tau \approx \Gamma^{-1} \cdot (1+s)^{-1/2}$. For the $Q$ parameter one thus obtains

$$Q_{\text{single atom}} \approx -\beta P_{out} \cdot (1 - g^{(2)}(t)) \cdot \tau = -\beta\frac{s}{2(1+s)^{3/2}},\tag{A.8}$$

with a minimum value of $Q_{min} = -0.19\beta$ at $s = 2$. For the source based on collective forward scattering, we calculate the same quantity. The exact temporal shape of the $g^{(2)}(t)$ function in the high-power limit is hard to calculate. Therefore, we make the simplifying assumption that for the considered power regime the only change of $g^{(2)}(t)$ is the reduction of the depth of the antibunching dip (see Fig. 8) while the overall temporal shape does not significantly change. That is, we approximate the temporal width to be independent of $s$ with $\tau_{coll} = 0.41\Gamma^{-1}$, which we obtain from the 2-photon theory prediction. In this way, we obtain for the saturation-dependent $Q$ parameter the solid blue curve shown in Fig. 9 which reaches a minimum value of $Q_{min} = -0.013$ at a saturation parameter of $s \approx 0.8$. For comparison, the solid gray curve is the Q parameter achievable for a perfect single photon source, i.e. a single atom with unit collection efficiency $\beta = 1$. In contrast, the dashed grey curve depicts the Q parameter of a single atom with a coupling strength of $\beta = 0.05$.

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
