# Peer review of "Higher-order mean-field theory of chiral waveguide QED"

_SciPost Physics Core, doi:SciPost Phys. Core 6, 041 (2023)_

## Round 1 · Referee Report · Anonymous (Referee 1) · 2022-9-21

Strengths

1 - timely topic at the forefront of a currently very active and rapidely developing field
2- addresses an important computational problem in quantum many body physics applying a well established method in a novel context allowing to reach a so far unexplored parameter regime
3-direct connection of the theoretical model with a concrete experimental setup demonstrating convincing agreement between measurements and theory predictions
4-nice and well structured introduction to the cumulant expansion method used
5- novel results and insight obtained by using and comparing various oders of the expansion model
6-experimental confirmation provides good evidence for the power and potential of the approach; this supports the use of the method in other less experimentally accessible cases

Weaknesses

1-technical quality of the figures and in particular the graphical presentation of the theoretical results is rather low
- poor choice of thin and many similar and hard to discern linetypes
-too many lines per plot
-too many and too tiny axis annotations
-too many plots/figure resulting in long and hard to read captions
2-description of the basic features of the experimental implementation of the system is rather condensed and compact; while this might be already available in other publications, a short review in the Appendix would be helpful to non-experts
3-the motivation for certain model assumptions could be improved by some extra comments and qualitative arguments for non experts:
-why are the invidual positions of the atoms appearing nowhere in the model ?
- why totally neglect backscattering / backward emission of photons; can the results be expected to be particuarly sensitive to this ?
- why are collective effects of the outside fiber loss/decay here completely negligible as atomic distances are of wavelength size (see for comparison NJP, 21(2), 025004, 2019) ?
-why are terms <sigma^x sigma^z> generally ignored in the expansion ?
- in Fig.6: what does a "mean-field" result for g2(0) represent ?
4-intro part to previous theoretical modeling misses a couple of papers

Report

Upon some improvements in presentation this work is very well suited for the journal

Requested changes

1 - improve/split figures
2-some more detailed review of experimental apparatus in Appendix
3-comment and clarify on some model assumptions for a broader readership
4-possible references to add in the intro:
Physical Review A 89.4 (2014): 043831
Physical Review A 96.3 (2017): 033857.

---

## Round 2 · Referee Report · Anonymous (Referee 1) · 2022-12-14

Strengths

see my first report

Weaknesses

1-technical quality of the figures and the graphical presentation is of sufficiently high quality now
2-description of the experimental implementation is improved
3-my other concerns here are largely answered satisfactorily

Report

Acceptance critera are now clearly met

Requested changes

remaining things to consider: while the dilute nature of the atoms along the fiber will reduce nearest neighbor shifts and interactions, collective effects via far field interference of the scattered light could still be important in the large atom number limit -<sigma_i^x sigma_i^z> ~ <sigma_i^y > terms should be kept in principle - Fig 8a: "mean-field" result for g2(0) ? Shouldn'd this be always equal to 1 as (a'a)^2 = (a')^2*(a)^2) in mean field ?

  • validity: high
  • significance: high
  • originality: high
  • clarity: high
  • formatting: good
  • grammar: excellent

Author:  Kasper Jan Kusmierek  on 2022-12-15  [id 3136]

(in reply to Report 1 on 2022-12-14)

We thank the referee for the positive feedback. Below we want to address the remaining questions:

  • While the dilute nature of the atoms along the fiber will reduce nearest neighbor shifts and interactions, collective effects via far field interference of the scattered light could still be important in the large atom number limit

We agree, that in principle collective effects may arise also due to decay to external modes. However, it is to be expected that these effects will be relatively small compared to the collective decay channels inside the waveguide on which we focused in the present work.

  • <sigma_i^x sigma_i^z> ~ <sigma_i^y > terms should be kept in principle

We agree on that point as well. Indeed, these terms are kept in our treatment. Products of Pauli operators referring to the same atom were always treated exactly by using the Pauli algebra. For example, we would write the above expression as <sigma_i^x sigma_i^z> ~ <sigma_i^y >=-i<sigma_i^y ><sigma_i^y >. Factorizations always refer to tensor products of Paulis referring to different atoms.

  • Fig 8a: "mean-field" result for g2(0) ? Shouldn'd this be always equal to 1 as (a†a)^2 = (a†)^2·(a)^2) in mean field ?

One has to differentiate on which level the mean-field approximation is applied. Throughout the work we only applied the cumulant expansion method on the level of atomic operators. For calculating g2, we first apply input-output relations to express g2 in terms of atomic correlations, which gives Eq. (28) of the manuscript. ’Mean field’ then means that we factorize all two-, three- and four-body correlation functions into products of single-particle averages. We emphasize that this is different from the factorization (a†a)2 = (a†)^2 · (a)^2 on the level of field operators. E.g. mean field on the level of atomic operators still allows for antibunching arising from each atom individually.

---

## Round 2 · Author Response

We thank the referee for the careful reading of the manuscript, the positive feedback, and the useful comments. Based on the listed weaknesses and requested changes, we edited the text and especially also the figures of our manuscript. In particular, we improved the style of presentation of figures and also reduced their number in order to keep a better focus on the main points of the manuscript. We hope the presentation is now clearer and the figures are easier to understand. The experimental apparatus and procedure is now described in more detail. Below we address the list of questions asked by the referee:

  • Q: Why are the invidual positions of the atoms appearing nowhere in the model?

The position of the atoms comes into play as a phase e^(±ikx_i) with a positive sign for absorption and a negative one for emission of photons. Thus, for forward scattering, and using the excitation laser as a phase reference, the phases cancel and the position of the atoms drops out. The position may still come in as a time delay between the atoms. In [1] it was shown that even in the non Markovian limit the time delay between emitters can be gauged away by redefinig the times for each consecutive system. In the Markovian limit, where the time delays are negligible, the distance between emitters amounts to an overall phase that can be gauged away by redefining the spin operators [2]. For this reason the positions of the atoms does not appear in the model.

  • Q: Why totally neglect backscattering / backward emission of photons; can the results be expected to be particuarly sensitive to this?

Back scattering can be largely suppressed in wave guide QED with chiral coupling. This motivates our consideration of the many-body dynamics in the ideal limit of exclusive forward scattering, realizing a cascaded quantum system. We believe it is natural and important to try to characterize and understand this limit first, before considering imperfections arising from back scattering. In the experiment, a certain level of back scattering is unavoidable. The fact the our theory compares very well to experimental data in many respects shows that the results tolerate a certain level of non-chirality. We also note that the inclusion of back scattering would complicate the theoretical model considerably, since the position of atoms would become relevant and, moreover, the cumulant expansion would become nonlinear. We also note that if the atoms are not ordered in a lattice which is commensurate with the wavelength of the input field the back-scattered light will not add up coherently. The backscattering-channel therefore plays a similar role as the other loss channels describing scattering out of the wave guide. The chiral arrangement with external loss channels is therefore a useful model for any ensemble of atoms that is not arranged into a lattice that is commensurate with the wavelength of the probe field.

  • Q: Why are collective effects of the outside fibre loss/decay here completely negligible as atomic distances are of wavelength size?

Even though the distance between neighboring lattice sites d = 447 nm is on the order of λ/2 = 426 nm, we expect that collective effects of the outside fiber loss/decay can be neglected. First, collective effects are very sensitive to the distance. In particular, the free-space coupling strength between atoms only becomes comparable to the free-space decay rate once the condition d ≤ λ/2π is reached (see Eq. (4a) in NJP, 21(2), 025004, 2019). Second, and more importantly, in spite of the close proximity of the trapping sites, the mean distance between two atoms is significantly larger than the wavelength due to the fact that the majority of the trapping sites remains empty. Concretely, in our experimental realization, we load ≈ 180 atoms over a length on the order of 1 mm into the optical lattice potential along the nanofiber. These atoms are thus distributed over ≈ 2, 000 lattice sites. Hence, we operate our system in a regime where collective effects from unguided modes can be neglected.

  • Q: Why are terms <sigma^x sigma^z> generally ignored in the expansion?

We know from equations (12) that moments <sigma^x_i> are zero, due to the resonant drive. Because of this, several terms in the equations of motion for <sigma^x sigma^y,z> cancel. The remaining terms describe a coupling among <sigma^x sigma^z> and <sigma^x sigma^y> only. Therefore, these quantities decouple from the dynamics and remain zero for all times. For this reason we left them out for sake of simplicity and readability. In our numerical calculations we take these terms into account and solve the complete set of equations without approximation (beyond the cumulant expansions). The numerical results confirm that the moments <sigma^x sigma^y,z> are zero on resonance.

  • Q: In Fig.6: what does a "mean-field" result for g2(0) represent?

We assume the referee was referring to Fig. 8 (Fig. 6 deals with squeezing spectra). Here, as well as regarding other observables, "mean-field" means truncation of the cumulant expansion at order 1, that is, higher order moments are factorized into products of first-order moments. For example, in equation (28), mean-field theory assumes <sigma^x_i sigma^y_j sigma^x_k sigma^y_l>=<sigma^x_i><sigma^y_j><sigma^x_k><sigma^y_l>. On the level of density matrices this assumes a tensor-product state.

[1] Gardiner, Phys. Rev. Lett. 70, 2269, 1993 [2] Pichler et al. , PRA 91, 042116, 2015

---

## Round 2 · List of Changes

• Fig. 2: Column corresponding to Pin/Psat=0.01 deleted and greater fontsize
  • Fig. 3: Spectra computed in mean-field theory deleted and greater fontsize
  • Fig. 4: y-axes in scientific notation; legend below plots; greater fontsize
  • Fig. 5: greater fontsize; line width increased
  • Fig. 6: greater fontsize
  • Fig. 7: Rows corresponding to <<sigma^y_i sigma^y_j>> and <<sigma^x_i sigma^x_j>> deleted; greater fontsize
  • Fig. 8: Plot correspondig to TO 3 deleted; greater fontsize
  • Fig. 9: Unified the x-axis of both plots; rearanged the plots in a column; greater fontsize; corrected minor mistake in legend
  • Fig. 10: greater fontsize; corrected minor mistake in y-axis label
  • Added Fig. 11 depicting the experimental setup
  • Corrected and completed the affiliations of authors
  • Added acknowledgements
  • Corrected equation (26a) for <<sigma^x_i sigma^x_j>>
  • Experimental setup more detailed explained
  • Added references

---

## Editorial Decision

published